# Emotional Eating Sustainability in Romania—A Questionnaire-Based Study

Anca Bacârea [1] , Vladimir Constantin Bacârea [2,*], Florin Buicu [3], Călin Crăciun [4], Bianca Kosovski [1], Raquel Guiné [5] and Monica Tarcea [6]

1 Department of Pathophysiology, University of Medicine, Pharmacy, Science and Technology "George Emil Palade", 540139 Targu Mures, Romania

2 Department of Scientific Research Methodology, University of Medicine, Pharmacy, Science, and Technology "George Emil Palade", 540139 Targu Mures, Romania

3 Department of Public Health, University of Medicine, Pharmacy, Science, and Technology "George Emil Palade", 540139 Targu Mures, Romania

4 Department of Surgery, University of Medicine, Pharmacy, Science and Technology "George Emil Palade", 540139 Targu Mures, Romania

5 CERNAS Research Centre, Polytechnic Institute of Viseu, 3504-510 Viseu, Portugal

6 Department of Community Nutrition and Food Safety, University of Medicine, Pharmacy, Science and Technology "George Emil Palade", 540139 Targu Mures, Romania

* Correspondence: vladimir.bacarea@umfst.ro

**Abstract:** Background: In Romania, there is little information regarding emotional eating and risk profile for obesity in eating disorders. Therefore, our purpose was to estimate the relationship between demographic, anthropometric, and emotional eating profiles among the Romanian adult population. Methods: The present study is descriptive and cross-sectional, involving 674 Romanian participants who answered the online questionnaire, which was developed and validated in 2019 through a European project. SPSS Statistics v.25 was used for statistical analyses, the Spearman test for linear regression, and Cronbach's alpha for the evaluation of the internal consistency of the scales. Results: The mean age of the studied population (mostly women, 67.95%) was $38.13 \pm 13.41$ years old, and the mean BMI (calculated based on self-declared weight and height) was $24.63 \pm 4.39$ kg/m$^2$; both measures are significantly higher in men than in women. BMI was also significantly higher in participants using food as an escape from situations such as stress, loneliness, feeling depressed, or as an emotional consolation. This behavior was observed especially in the elderly, similar to other European countries. Conclusion: Our data contribute to a better understanding of emotional eating in Romania, and we hope to improve public health policies, with the goal of preventing obesity and chronic related disorders.

**Keywords:** emotional eating; BMI; food behavior; obesity



## 1. Introduction

Eating behavior has been a subject of study for a long time. Different factors such as socio-demographic, cultural, economic, emotional, and environmental factors were evaluated [1,2]. One of our previous studies focused on Romanian people's motivations toward healthy eating [3]. It is a fact nowadays that people no longer consume food merely to satisfy metabolic needs and satiety [4], but also as a reaction to emotions, which in turn are influenced by biological, social, and cultural factors as well [5,6].

The concept of emotional eating has appeared, and it is traditionally defined as eating in response to negative emotions [7] and, more broadly, not only as a response to negative emotions but also as an ego-threat or distress response [8,9]. A study indicates that not only eating in response to negative emotions should be considered, but also eating in response to positive emotions, highlighting the relationship between eating and mood

amelioration [10]. A meta-analysis indicated that positive emotion resulted in increased eating and that restrained eaters are vulnerable to emotion-induced eating [11]. Overeating in response to emotions was associated with increased body mass index (BMI), overweight, and obesity [9,12]. Different theories such as the psychosomatic theory (eating may reduce anxiety or distress), the externality theory (excessive reaction to external food stimuli), and the restriction theory (the disinhibitors cause people who chronically restrict their food intake to overeat) were associated with overeating and obesity [12–14]. Some disorders (e.g., diabetes, cardiovascular diseases, respiratory disorders, neurodegenerative conditions, psychiatric disorders, and cancer) have a high risk of occurrence when obesity is present [15]. Obesity also has an effect on fetal development, birth outcomes, and child health [16].

In the context of nutrition, the term "mindful eating" means "paying attention in a particular way, on purpose, in the present moment, and nonjudgmentally" [17] to food and to the physical and emotional experience of food [18]. The practice of mindful eating is included by nutritionists in their strategy to change the general approach to eating, as nutritionists are aware that diet alone is not effective. In this way, people learn to change their food-related behavior, as they recognize and respond to satiety, and not respond to inappropriate stimuli for eating, such as nutrition-related mass-media publicity, boredom, or emotions [18,19]. The results of one study aimed to evaluate the impact of mindful eating on eating behavior in overweight and obese women, and these results showed that both emotion dysregulation and negative affect are associated with greater emotional eating [20].

The COVID-19 pandemic contributed to accentuating the phenomenon of emotional eating, as many socio-economic, cultural, and lifestyle changes occurred, and interpersonal relationships were affected worldwide. One Romanian study reports that a significant number of normal-weight participants, as well as overweight and obese participants, gained weight [21]. Many other studies provide evidence of the negative effects of isolation and lockdown on emotional well-being and eating behavior [22–24].

Being overweight or obese are major health issues worldwide, and they are equally problematic in Romania. According to one study, the prevalence of overweight and obesity in Romania was 31.1% and 21.3%, respectively [25]. According to the World Health Organization (WHO) 2022 report, the 2016 prevalence of overweight (including obesity) among Romanian adults was 57.7% for both sexes (64.3 in men and 51.1 in women) [26].

The phenomenon of emotional eating has been little studied in Romania and is usually in association with certain pathologies or in narrow categories of the population [12,27].

The aim of our study was to assess demographic, anthropometric data and emotional eating profiles in a Romanian population cohort and, therefore, to contribute to a better understanding of the emotional eating phenomenon with the potential to improve the prevention strategies and to reduce the burden of overweight and obesity. This study is a part of a multinational project that was developed in 2018–2020 and that was titled "Psycho-social motivations associated with food choices and eating practices (EATMOT)", in which 16 countries were included (Argentina, Brazil, Croatia, Cyprus, Egypt, Greece, Hungary, Italy, Latvia, Lithuania, Macedonia, Netherlands, Poland, Romania, Serbia, Slovenia, and The Unites States of America).

## 2. Materials and Methods

The current study is a descriptive cross-sectional questionnaire-based study with the goal of evaluating emotional eating profiles based on the answers of 674 Romanian adults who completed the online questionnaire. The questionnaire, which was developed and validated within the EATMOT project by Ferrão et al. [28], was translated into the Romanian language. The translation and back translation phases were performed by separate and independent teams of experts. The study is country-representative, as the participants are from different regions of Romania. The study was approved by the Ethics Committee of the University of Medicine and Pharmacy, Science, and Technology "G.E. Palade" from Targu Mures and was conducted in accordance with the Helsinki Declaration.

The participants answered questions that referred to demographic data, anthropometric data, and questions about emotional eating aspects. The investigated parameters were as follows: age; gender; environment (urban, rural); weight; height; and current employment status (student, employed, unemployed, retired). In terms of age, the participants were classified into five categories as follows: 18–29, 30–39, 40–49, 50–59, $\geq$60 years old.

BMI was calculated based on self-declared weight and height ($kg/m^2$).

The following questions were used to investigate emotional eating:

Q1—Food helps me cope with stress.
Q2—I usually eat food that helps me control my weight.
Q3—I often consume foods that keep me awake and alert (such as coffee, coke, and energy drinks).
Q4—I often consume foods that help me relax (such as some teas, and red wine).
Q5—Food makes me feel good.
Q6—When I feel lonely, I console myself by eating.
Q7—I eat more when I have nothing to do.
Q8—For me, food serves as an emotional consolation.
Q9—I have more cravings for sweets when I am depressed.

The study participants were able to answer the nine questions with the following choices: totally disagreed, disagree, neither agree nor disagree, agree, or totally agree. We did two composite scales: block 1, including questions Q1, Q6, Q8, and Q9 to investigate food as an escape, and block 2, including questions Q4 and Q5 to investigate food associated with well-being. Questions Q2, Q3, and Q7 were left to stand alone.

Data collection was performed using MS EXCEL. For the data analysis, we used Graph Pad Prism (demo version), Epi Info 7, and SPSS Statistics v. 25. For the quantification of the variables, the mean and DS, or the median and the values were calculated according to the case (normal distribution or not, continuous variables, or discrete variables) and extremes (range). Spearman linear regression and Pearson's correlation coefficient were used to determine relationships between variables. The interpretation was made as follows: not existing ($r = 0$), very weak ($0.00 < r < 0.10$), weak ($0.10 \leq r < 0.30$), moderate ($0.30 \leq r < 0.50$), strong ($0.50 \leq r < 0.70$), very strong ($0.70 \leq r < 1$), or perfect ($r = 1$) [28,29]. Cronbach's alpha coefficient was used to determine the internal consistency of responses. The interpretation was as follows: over 0.9: excellent; 0.8–0.9: very good; 0.7–0.8: good; 0.6–0.7: medium; 0.5–0.6: reasonable; below 0.5: bad [30]. The Chi test was used to establish statistical significance in the case of categorical variables. A *p*-value below 0.05 was considered statistically significant.

## 3. Results

The socio-demographic parameters (age, gender, environment, and employee status) of the studied population are presented in Table 1.

It can be observed that most participants in the study are part of the age group of 18–29 years (young adults), are females (the number of women is twice that of men), come from urban areas, and are employed. The mean age of the studied population was $38.13 \pm 13.41$ years old (min 18 years; max 80 years). The mean age of women was $36.60 \pm 0.56$ years, and that of men was $41.05 \pm 0.91$ years, which is a significant difference ($p < 0.0001$).

The mean BMI was $24.63 \pm 4.39$ $kg/m^2$ (min 16; max 48). The mean BMI of men ($25.89 \pm 0.25$ $kg/m^2$) was significantly higher than the mean BMI of women ($23.95 \pm 0.19$ $kg/m^2$) ($p < 0.0001$).

We observed a slow increase in BMI value ($r^2 = 0.1325$, $p < 0.0001$) with aging.

Table 2 shows the frequency of participants' answers to the nine questions investigating emotional eating.

**Table 1.** The socio-demographic data that characterize the studied population.

| Parameter | No [1] | % | CI [2] (%) |
|---|---|---|---|
| Age | | | |
| 18–29 yo | 227 | 33.70 | 30.1–37.4 |
| 30–39 yo | 119 | 17.70 | 14.9–20.8 |
| 40–49 yo | 164 | 24.30 | 21.2–27.8 |
| 50–59 yo | 122 | 18.10 | 15.3–21.3 |
| ≥60 yo | 42 | 6.20 | 4.6–8.4 |
| Gender | | | |
| Female | 458 | 68.00 | 64.3–71.4 |
| Male | 216 | 32.00 | 28.6–35.7 |
| Environment | | | |
| Urban | 562 | 83.40 | 80.3–86.1 |
| Suburban | 19 | 2.80 | 1.8–4.4 |
| Rural | 93 | 13.80 | 11.3–16.7 |
| Employee status | | | |
| Student | 127 | 18.80 | 16.0–22.0 |
| Employed | 490 | 72.70 | 69.1–76.0 |
| Unemployed | 32 | 4.70 | 3.2–6.5 |
| Retired | 25 | 3.70 | 2.5–5.5 |

[1] No = number of participants; [2] CI = confidence interval.

**Table 2.** Frequency of participants' answers to emotional eating questions.

| Question | 1 no (%) | 2 no (%) | 3 no (%) | 4 no (%) | 5 no (%) |
|---|---|---|---|---|---|
| Q1 | 88 (13.10) | 161 (23.90) | 196 (29.10) | 187 (27.70) | 42 (6.20) |
| Q2 | 26 (3.90) | 96 (14.20) | 239 (35.50) | 270 (40.10) | 43 (6.40) |
| Q3 | 76 (11.30) | 130 (19.30) | 159 (23.60) | 160 (23.70) | 149 (22.10) |
| Q4 | 19 (2.80) | 84 (12.50) | 307 (45.50) | 217 (32.20) | 47 (7.00) |
| Q5 | 25 (3.70) | 79 (11.70) | 234 (34.70) | 286 (42.40) | 50 (7.40) |
| Q6 | 158 (23.40) | 205 (30.40) | 140 (20.80) | 146 (21.70) | 25 (3.70) |
| Q7 | 108 (16.00) | 137 (20.30) | 137 (20.30) | 249 (36.90) | 43 (6.40) |
| Q8 | 201 (29.80) | 187 (27.70) | 133 (19.70) | 136 (20.20) | 17 (2.50) |
| Q9 | 122 (18.10) | 148 (22.00) | 155 (23.00) | 195 (28.90) | 54 (8.00) |

1—totally disagree, 2—disagree, 3—neither agree nor disagree, 4—agree, 5—totally agree.

The highest frequencies of answer no. 3 (neither agree nor disagree) were recorded for Q1 and Q4.

Table 3 indicates item–item correlations for the questions included in block 1 and aims to evaluate food as an escape. Based on the value of the Cronbach alpha of 0.868, which is very good, we accepted the questions in the composite scale.

Table 4 indicates item–item correlations for the questions included in block 2 and aims to evaluate the association between food and well-being. Based on the value of the Cronbach alpha of 0.682, which is a medium value, we accepted the questions in composite scale.

**Table 3.** Item–item correlations for questions contained in block 1 evaluating food as an escape (block 1) [1].

| Item | Q1 | Q6 | Q8 | Q9 |
|---|---|---|---|---|
| Q1 | 1 | | | |
| Q6 | 0.555 ** | 1 | | |
| Q8 | 0.576 ** | 0.847 ** | 1 | |
| Q9 | 0.480 ** | 0.627 ** | 0.646 ** | 1 |

[1] Cronbach alpha = 0.868, ** Correlation is significant at the 0.01 level (2-tailed).

**Table 4.** Item–item correlations for questions contained in block 2, which evaluated food and well-being [1].

| Item | Q4 | Q5 |
|---|---|---|
| Q4 | 1 | |
| Q5 | 0.518 ** | 1 |

[1] Cronbach alpha = 0.682, ** Correlation is significant at the 0.01 level (2-tailed).

We assessed the associations between the questions contained in block 1, block 2, and questions left alone (Q2, Q3, and Q7) and the studied parameters. The results are presented in Table 5.

**Table 5.** Associations between the questions contained in block 1, block 2, and questions left alone (Q2, Q3, and Q7) and the studied parameters (*p*-value 1).

| Parameter | Block 1 | Block 2 | Q2 | Q3 | Q7 |
|---|---|---|---|---|---|
| Age [2] | $X2$ (16, No = 674) = 50.65 $p = 0.0000$ | $X2$ (16, No = 674) = 31.31 $p = 0.0123$ | $X2$ (16, No = 674) = 33.90 $p = 0.0056$ | $X2$ (16, No = 674) = 36.92 $p = 0.0021$ | $X2$ (16, No = 674) = 37.00 $p = 0.0021$ |
| Gender [2] | $X2$ (4, No = 674) = 27.12 $p = 0.0000$ | $X2$ (4, No = 674) = 20.10 $p = 0.0050$ | $X2$ (4, No = 674) = 1.24 $p = 0.8711$ | $X2$ (4, No = 674) = 35.17 $p = 0.0000$ | $X2$ (4, No = 674) = 10.02 $p = 0.040$ |
| Environment [2] | $X2$ (8, No = 674) = 7.65 $p = 0.4678$ | $X2$ (8, No = 674) = 4.33 $p = 0.8258$ | $X2$ (8, No = 674) = 10.74 $p = 0.2167$ | $X2$ (8, No = 674) = 6.34 $p = 0.6088$ | $X2$ (8, No = 674) = 7.67 $p = 0.4657$ |
| Employee status [2] | $X2$ (16, No = 674) = 23.32 $p = 0.1054$ | $X2$ (16, No = 674) = 18.46 $p = 0.2972$ | $X2$ (16, No = 674) = 25.72 $p = 0.0580$ | $X2$ (16, No = 674) = 26.90 $p = 0.0425$ | $X2$ (16, No = 674) = 26.52 $p = 0.0470$ |
| BMI [3] | r = 0.2464 $p = 0.0001$ | r = 0.0310 $p = 0.4206$ | r = 0.0632 $p = 0.1007$ | r = 0.1551 $p = 0.0001$ | r = 0.2001 $p = 0.0001$ |

$p < 0.05$ was considered significant; [2] Chi square test for n x m table; [3] Spearman test.

### 3.1. Age

In the age category ≥60 years old (elderly group), we registered the highest frequency of answers "agree" to the questions contained in block 1. Regarding questions contained in block 2, both young adults and the elderly responded "agree" and "totally agree".

Choosing food in order to control weight is a concern, especially for the elderly. The consumption of stimulant foods was associated with the 30–59 age group (Q3). In the elderly group, we obtained a significantly higher number of answers "agree" and "totally agree" to Q7, which means that they eat more when they have nothing to do.

### 3.2. Gender

Men answered mostly "agree" and "totally agree" to the questions contained in blocks. Women consumed more stimulating foods (Q3). Women responded more with the answer "totally agree" to Q7.

### 3.3. Environment

The environment of origin did not influence the answer to the questions.

### 3.4. Employee Status

There are no significant differences with regard to professional status between the answers to the questions in block 1 and 2. However, the answer to the questions in block 2 for most of the retired participants was "agree". The retired participants responded more with the answers "agree" and "totally agree" to Q2, although this difference was statistically insignificant. The employed and unemployed participants answered mostly "agree" and "totally agree" in Q3. The retired participants answered mostly "agree" and "totally agree" to Q7.

### 3.5. BMI

People who agreed and totally agreed with questions included in block 1 had a significantly higher BMI. No significant association was observed when investigating BMI in relation to questions included in block 2 or in relation to Q2. Although significant in the case of Q3 and Q7, the low values of r do not allow us to establish any associations.

## 4. Discussion

Emotional status influences all aspects of human life, from using emoticons in response to messages to eating behavior. Some people tend to choose food according to their emotions and mood, and a common reaction is to overeat in order to compensate for negative or positive emotions [4,10,31] and the further occurrence of obesity. Being overweight and being obese represent a growing problem in Romania, as shown in the introduction. In a previous study, we observed a significant positive association between BMI and glycemia in adults older than 22 years old [32]. Knowing the link between obesity and the risk of occurrence of many diseases, we consider it to be of high importance to explore emotional eating in Romania, a country where little research has been conducted in this regard.

The highest percentage of responses recorded by us was in the 18–29 age group, and a relatively low number of participants were over 60 years old. This indicates on the one hand that the results are relevant for the young population from the perspective of future public health policies, and on the other hand, it indicates the need to continue the study and include more elderly people. Many studies focus on emotional eating in young adults [33,34], and this is a gap that must be filled.

The questionnaire we used includes questions investigating emotional eating as a response to emotions (stress, feeling depressed, loneliness) or as an emotional consolation (Q1, Q6, Q8, and Q9), due to boredom (Q7), for well-being (Q4 and Q5), because of the need to stay alert (Q3), or from a conscious choice of food (Q2).

There was no difference between young and elderly adults regarding food and well-being, as the results show that no matter the age, consuming coffee, a glass of wine, or some comforting food helps people to relax. Our results show that especially elderly people associate food with an escape behavior, in contrast with the results of another study [35]. This discrepancy could be due to the mean age value of 38 years that we obtained in our group versus that of 50 years in the mentioned study.

Naturally, the active population and especially women use more stimulants to be alert and to manage their daily activities. The elderly declared that they are more interested in choosing a diet that will maintain their weight. Given that with aging, BMI increases [3,36] and different pathologies are usually present, this is the right thing to do; however, if we look at Q7, we do not find the same choice, as elderly participants declared more frequently that they eat when they have nothing to do. One study aiming to evaluate the relationships between eating behaviors and food cravings, divides the participants into two age categories (≤25 years versus >25 years) and indicates an association between increased cognitive restraint and decreased food cravings in the ≤25 years group (36). This study

indicates that eating behaviors or food cravings are not influenced by gender [36], whereas other studies indicate that women in particular have such behaviors [36,37].

Our results indicate that men in particular use food as an escape, and, indeed, they have a significantly higher BMI than women. However, this finding is contradicted by the results of other studies [38,39] and, therefore, must be further investigated on a larger population of men, because the number of male participants in this study was much lower than the number of female participants (216 versus 458), and also because the mean age of men was significantly higher than the mean age of women. According to Nolan et al., there was no sex difference for negative emotions regarding the tendency to eat more, but men reported a tendency to eat more when they experienced positive emotions [9].

Regarding the employment status, the retired participants, who are usually elderly persons, answered mostly "agree" and "totally agree" to Q7, as they have more free time. This fact brings boredom into the discussion. One study using an overweight/obese sample population showed that boredom was related to poorer psychological well-being and difficulties with emotional regulation [40]. Another study using a sample of college students indicated that inadequate eating behavior in response to boredom, negative affect, or external stimuli is very likely to occur in those prone to boredom and difficulties in emotional regulation [33]. According to other studies, boredom, proneness, and interoceptive ability are important goals in the prevention and treatment of emotional eating and should be considered a separate dimension of emotional eating [41,42].

According to our results, weight gain was associated with block 1, but not with block 2 nor with Q2. This is consistent with the results of other studies, suggesting that weight gain and obesity can be caused by the tendency to eat more in response to negative emotions [9,43–46]. Emotional eaters can respond both to negative and positive emotions, although some results indicate weight gain only in association with negative emotions [40,47] or with female gender only [37].

People who choose a food that helps them maintain their weight do not tend to increase their BMI, indicating that they are self-conscious about their weight and health. High self-control was associated with a low increase in overeating behaviors and BMI and an important improvement in healthy diet over time [48].

A beneficial association for emotional eaters is physical activity, according to one study's results, because despite their need to eat when under emotional distress, they make healthier food choices to cope with this distress [49,50].

Our study has some limitations because of the small number of men compared to that of women and the small number of elderly people included in the study. As data were self-reported by the participants, there is a risk that emotional eating could actually be low self-control or, if especially emotional eaters overestimate their food intake, because of distress [7,51]. We hope that our data will contribute to a better understanding of emotional eating in Romania and will improve public health policies, with the goal of preventing obesity and chronic related disorders.

## 5. Conclusions

Emotional status and environmentally sensitive exposures influence all aspects of human life, especially eating behavior. It is necessary to target the vulnerable populational groups that have the tendency to choose food according to their emotions and mood, followed by overeating/undereating in order to compensate for negative or positive emotions and the further occurrence of obesity or eating disorders.

To build the capacity to manage this addictive food behavior, it is important to evaluate the risk profiles for vulnerable groups based on their attitudes and knowledge toward nutrition, food culture, accessibility to food products, sustainable resources, health status, and psychological characteristics as well.

The main findings of this study were the following: BMI was significantly higher in participants who agreed and totally agreed to the variables about food as an escape, and the risk profiles were higher for the female gender and adult working Romanian

participants, meaning that food-related-problem management requires more focus on multidisciplinary community interventions based on screening, specific national programs, and better communication skills for professionals in order to obtain better outcomes.

**Author Contributions:** Conceptualization: A.B. and M.T.; methodology: A.B. and V.C.B.; software: V.C.B.; validation: A.B. and V.C.B.; formal analysis: A.B. and V.C.B.; investigation: A.B., V.C.B., F.B., C.C., B.K. and M.T.; resources: M.T.; data curation: A.B. and V.C.B.; writing—original draft preparation: A.B.; writing—review and editing: A.B., V.C.B., F.B., C.C., B.K., R.G. and M.T.; visualization: A.B. supervision: A.B. and M.T.; project administration: R.G. and M.T.; funding acquisition: R.G. and M.T. All authors have read and agreed to the published version of the manuscript.

**Funding:** This research was funded by the Research Center CI&DETS (Polytechnic Institute of Viseu, Portugal) with grant n.º PROJ/CI&DETS/CGD/0012. The APC was funded by FCT-Foundation for Science and Technology (Portugal), scholarship number UIDB/00681/2020.

**Institutional Review Board Statement:** The study was conducted in accordance with the Declaration of Helsinki, and approved by the Ethics Committee of Politechnic Institute Of Viseu—School Of Health, Portugal no.04/2 February 2017. The agreement for the project implementation from the of University of Medicine, Pharmacy, Sciences and Technology "George Emil Palade" was released in 22 June 2016, no. 8892.

**Informed Consent Statement:** Informed consent was obtained from all subjects involved in the study.

**Data Availability Statement:** Data are available from the corresponding author upon reasonable request.

**Acknowledgments:** This work was prepared in the ambit of the multinational project EATMOT from CI&DETS Research Centre (IPV—Viseu, Portugal) with reference PROJ/CI&DETS/CGD/0012. This work is supported by Portuguese National Funds through the FCT—Foundation for Science and Technology, I.P., within the scope of the project Refª UIDB/00681/2020. Furthermore, we would like to thank the CERNAS Research Centre and the Polytechnic Institute of Viseu for their support.

**Conflicts of Interest:** The authors declare no conflict of interest.

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
