# Peer review of "Emotional Eating Sustainability in Romania—A Questionnaire-Based Study"

_sustainability, doi:10.3390/su15042895_

Round 1
Reviewer 1 Report
In this paper, the authors aimed to assess demographic, anthropometric data, and emotional eating profiles, in a Romanian population cohort.
The topic is interesting and the results have the potential to improve the prevention strategies for overweight and obesity.
Major comments
First of all, the purpose of this research is not clear. In the method section, the authors declared that they used a questionnaire developed and validated within the EATMOT project by Ferrão et al. (28) was translated into the Romanian language. That questionnaire analyzed for internal validation of the questionnaire, in terms of semantics and internal structure, for the Portuguese population, so that it could then be used by the participants in other countries in a later phase of the project. Was the current study conducted to attest to the validity of the emotional eating scale in Romania?
if in the present study, the questionnaire was applied without being validated for semantics in the Romanian language, what exactly was the purpose of the study? What was the value of Cronbach's alpha for the application of the 10 items in the population studied from Romania?
The authors also declared that ”the study is country-representative as the participants are from different regions of the country but it was not explained how they reached this conclusion (percentage of participants from each region?).
The item scores were not calculated and the result of the investigations is not clear.
The statistical analysis is not clear.
Minor comments:
It is not clear what young and elderly adults mean.
Author Response
Dear Reviewer,
Thank you for evaluating the article entitled “Emotional Eating sustainability in Romania - a question-naire-based study”, authored by Anca Bacârea , Vladimir Constantin Bacârea, Florin Buicu, Călin Crăciun, Bianca Kosovski, Raquel Guiné, Monica Tarcea.
We have tried to answer your requirements, as follows:
- The purpose of this study was to describe the emotional behavior related to food in Romania and not to validate the questionnaire. The questionnaire was not only applied in Romania, but in other 16 countries simultaneously. In this process the translation followed back-translation process, based on principles that attest the validity of the translation process into each of the 16 native languages. This means that the translation and back translations phases were done by separate and independent teams of experts, including language, precisely to avoid any kind of misinterpretation. This and other principles of the back translation procedure ensure validity is kept across different languages. We added this information to the material and methods section.
- The calculation of alpha is not to validate the semantics of the questions, but measures internal reliability of the scale. We considered that the nine questions answer different issues related to emotional eating, which is why they were analyzed either in blocks or individually. The Cronbach test was applied to establish the internal validity of the answers in the case of blocks.
- The questionnaire was promoted online through the university centers in Romania, which are spread over the entire surface of the country, and in each historical region of Romania. That is why we considered that the study is relevant for the whole country.
- We consider that the statistics is adequate. We kindly ask you to specify where it is unclear so we can correct it;
- In the text, lines 150-151, it is specified that the age category 18-29 years refers to young adults; those in the age category ≥ 60 years were considered in the elderly group and we added this information in line 185.
Hoping our work fits your expectations,
Kind regards,
The authors

Reviewer 2 Report
In my opinion, the study design and concept are very good. The aim of the study was clearly defined. I have a few comments and suggestions regarding to the methods, results and discussion section.
Introduction
Based on the new published statistics of incidence and prevalence of overweight and obesity according to WHO from 2022, consider updating the data in line no. 79. Consider removing the last sentence in lines 81-83 (taken out of context).
Materials and Methods
In the "Results" section, you mention the Chí-test, but there is no mention of it in the methodology. Add it to the methodology.
Results
Table 1 - in the table, the label "age" is inappropriately shifted. Is it necessary to indicate CI for data indicating age, gender, environment and employee status?
The passage in lines 246-247 is not supported by any available data - complete the table comparing the differences between men and women to make it clear on what basis you are making these statements.
Discussion
Consider removing / moving to another part of the manuscript the passage in lines 215-218.
Finally, I recommend giving the authors a chance to revise the manuscript after minor revision.
Author Response
Dear Reviewer,
Thank you for evaluating the article entitled “Emotional Eating sustainability in Romania - a question-naire-based study”, authored by Anca Bacârea , Vladimir Constantin Bacârea, Florin Buicu, Călin Crăciun, Bianca Kosovski, Raquel Guiné, Monica Tarcea.
We have tried to answer all your requirements, as follows:
- We have completed the document with data regarding the prevalence of overweight and obesity according to WHO from 2022;
- We removed the sentence in lines 81-83;
- We added the Chi test to the statistics section;
- We moved the age parameter, along with the other parameters; it is usual from the point of view of descriptive statistics to calculate the CI as an indicator of the distribution of If you think they are extra, we can delete them.
- In the Results section, lines 149-151, the average BMI values for men and women are calculated, as well as the statistically significant p value; due to the format problems of table 1, we consider that the information can remain in the text;
- We removed passage in lines 215-218.
Hoping our work fits your expectations,
Kind regards,
The authors

Round 2
Reviewer 1 Report
I agree with the publication of the article in its present form.